# Imaging Techniques and Clinical Application of the Marrow–Blood Barrier in Hematological Malignancies

**DOI:** 10.3390/diagnostics14010018

**Published:** 2023-12-21

**Authors:** Jianling Zhang, Qianqian Huang, Wenjin Bian, Jun Wang, Haonan Guan, Jinliang Niu

**Affiliations:** 1Department of Medical Imaging, Shanxi Medical University, 56 Xinjian South Road, Taiyuan 030001, China; niaiiiii@163.com (J.Z.); qianqianhuang77@163.com (Q.H.); mribianwenjin@163.com (W.B.); 2Department of Radiology, The Second Hospital of Shanxi Medical University, No. 382 Wuyi Road, Taiyuan 030001, China; cjr.wangjun@vip.163.com; 3MR Research China, GE Healthcare, Beijing 100176, China; haonan.guan@ge.com

**Keywords:** marrow–blood barrier, imaging techniques, clinical application, MBB dysfunctions

## Abstract

The pathways through which mature blood cells in the bone marrow (BM) enter the blood stream and exit the BM, hematopoietic stem cells in the peripheral blood return to the BM, and other substances exit the BM are referred to as the marrow–blood barrier (MBB). This barrier plays an important role in the restrictive sequestration of blood cells, the release of mature blood cells, and the entry and exit of particulate matter. In some blood diseases and tumors, the presence of immature cells in the blood suggests that the MBB is damaged, mainly manifesting as increased permeability, especially in angiogenesis. Some imaging methods have been used to monitor the integrity and permeability of the MBB, such as DCE-MRI, IVIM, ASL, BOLD-MRI, and microfluidic devices, which contribute to understanding the process of related diseases and developing appropriate treatment options. In this review, we briefly introduce the theory of MBB imaging modalities along with their clinical applications.

## 1. Introduction

Human blood cells are produced in the extravascular space of the BM, and to enter the circulation, blood cells must migrate to the barrier between the hematopoietic compartment and the circulation, the MBB [1]. Several observations have demonstrated that MBB regulates cell trafficking by retaining immature or defective cells while permitting the passage of suitable cells [1,2]. Different blood cells migrate in different ways. Red blood cells pass through the sinus endothelial cell pores by increasing pressure. Leukocytes first adhere to the sinus wall and then pass through the endothelial cells through cellular deformation. Immature hematopoietic cells generally cannot pass through the sinus wall, and even mature blood cells do not all enter the circulation. The sinus wall barrier makes a large number of blood cells stored in the bone marrow, which can be released by regulation for the body under a stress state. In the case of injury, such as tumors, the structure of the blood sinus wall becomes incomplete, and most of the sinus wall has only a single layer of endothelial cells, so that a large number of mature red blood cells and even immature cells pass through the sinus wall into the circulation. According to a previous report, dysfunction of the MBB may manifest as increased barrier permeability and angiogenesis, as patients with hematological diseases such as acute leukemia and preleukemia diseases had higher levels of angiogenesis in the BM [3]. The study showed that the increased endothelial permeability of the blood sinus wall in acute myeloid leukemia (AML) patients makes it easy for leukemia cells to hide, escape chemotherapy and enter the blood sinus, which is one of the factors of AML recurrence. Therefore, evaluating the functional status of MBB is essential for understanding pathogenesis and developing therapeutic strategies for related diseases. In the 20th century, scanning electron microscopy (SEM) or transmission electron microscopy (TEM) were used to observe the structural characteristics of the MBB and blood cell transmigration into circulation, but these methods were invasive and could not continuously observe the changes in the MBB. Dynamic contrast-enhanced MRI (DCE-MRI), intravoxel incoherent motion (IVIM), arterial spin labeling (ASL), and BOLD-MRI are commonly utilized to monitor MBB function by providing information on perfusion characteristics, micro-vessel density (MVD), and permeability assessment. These methods are non-invasive and can accurately reflect the function of the MBB, which can be applied to the diagnosis and prognosis of diseases in the future. In the meantime, microfluidic devices can mimic the physiological characteristics of the microenvironment, enabling direct assessment of both integrity and endothelial sinus permeability within the MBB [4]. In this review, we introduce MBB assessment methods and discuss their clinical applications and the advantages and disadvantages of each approach.

## 2. Cellular Components and Function of the MBB

The structure of the MBB is mainly composed of the blood sinus wall (endothelium, adventitial cells, and basement membrane) (Table 1) [5]. Previous studies of the MBB have required in vivo experiments, so the understanding of the MBB lags behind that of other areas of hematopoiesis. Since PEASE (1955) reported the ultrastructure of the BM, a large amount of data on the fine structure of the BM sinus has been accumulated [6]. The sinus is thin and, in many areas, consists of only a single continuous layer of flat endothelial cells, often with little subendothelial connective tissue and no obvious basement membrane [7].

The endothelial cells are located on the medial surface of the medullary sinus, and a large number of fenestrae make the medullary sinus communicate with the hematopoietic tissues outside the medullary sinus. Fenestrated structures within vascular endothelial cells in the medullary sinus possess the ability to expand, contract, or close [8], thereby influencing both the number and timing of mature blood cells released from the hematopoietic zone into the blood circulation. Under TEM, endothelial cells were found to be thin, and the attenuated cytoplasm varied in thickness and was occasionally interrupted by fenestrations of different sizes. Under SEM, the fenestrations can be divided into two categories by size: one category comprises larger fenestrations that allow for the migration of blood cells, while another category consists of smaller pores forming a cribriform plate through aggregation [9].Adventitial cells reside in the outermost layer of the medullary sinus, wrapping vascular endothelial cells and covering two-thirds of the surface area of vascular endothelial cells. Adventitial cells regulate the area covering vascular endothelial cells and control the fenestra size by shrinking the cell body and stretching the surface processes of vascular endothelial cells, which can cooperate with vascular endothelial cells to regulate the volume of the medullary sinus and the flow of blood cells into and out of the hematopoietic area [10,11]. Elongated cytoplasmic processes provide a framework for extravascular tissue [12].The basement membrane is composed primarily of laminin and a small amount of type II collagen, which are non-continuously distributed on the outer surface of vascular endothelial cells in the medullary sinus, as well as between reticular cells and vascular endothelial cells in the outer membrane, and can effectively regulate the exchange and entry of chemicals and cells inside and outside the medullary sinus.

### 2.1. Normal MBB

The endothelial cells of the MBB appear to be the structural basis for the “selectivity” of normal the MBB in translating a signal that expresses the body’s need for the cell type to be delivered [1]. Blood cells enter the circulation from the medullary cord not at random locations in the sinus wall but by forming a hole near it. This may be because cells will flow out of the site of least resistance, where endothelial cells are passive. The migrating cells create contact between the basement membrane and the luminal endothelial plasma membrane, inducing membrane fusion and pore formation, which closes once the cells have passed. The presence of adjacent dense endothelial cell junctions may provide support for the endothelial cell edge, thereby facilitating cytoplasmic perforation [4]. Mature cells have an enhanced ability to “drill” holes compared to immature cells, as confirmed by the dense migration pores and clearly deformed migrating cells; thus, the release of mature cells is selective [13,14,15].

In addition to cells, the MBB can selectively transport some physiological molecules (the molecules of substances with physiological functions, such as the iron–transferrin complex). The difference in the concentration of intramembranous protein particles between the luminal and tissue sides of the sinus epithelium is thought to be related to the direction of transport [16]. It may be transported via the intracellular route of endocytosis [17], where endocytic vesicles can either fuse into storage vesicles to sequester material or form transfer tubules to process material [18]. For example, the iron–transferrin complex cannot cross the intercellular junction, but can be mediated by vesicular transport, uptake of particulate matter from the inside to the outside of the endothelium.

### 2.2. Abnormal MBB

In BM-infiltrative diseases, both neoplastic and non-neoplastic cases can present with naive leukocyte hemogram (referring to the presence of immature white blood cells in the peripheral blood), indicating impaired function of the MBB [19]. Hyperleukocytosis occurs in 5–20% of untreated AML patients, and its mechanisms are related not only to the unlimited proliferation of leukemia cells but also to the severe damage to the MBB. Cell proliferation is the process by which cells grow and divide to produce two daughter cells. The process of cell proliferation is also regulated by a certain mechanism in the body. Once this regulation is lost, it will lead to unlimited proliferation of cells, loss of physiological function of cells, and abnormal morphology, which is a cause of cancer. It has been shown that leukemic cells first proliferate in the hematopoietic compartment and then invade the sinusoids. The first element of sinus wall elimination is the basal material of the outer membrane layer and basement membrane, leading to the appearance of large gaps or areas of damage in the endothelium. Finally, the sinuses disappear and the BM is filled with leukemic cells. Destruction of the sinusoid is likely to interfere with the release of leukemic cells and aggravate the already severe BM infiltration [20]. The results of transmission electron microscopy showed that the number of membrane pores of sinus endothelial cells in acute and chronic myelogenous leukemia patients was significantly higher than that in normal controls, and the coverage rate of sinus endothelial cells in leukemia patients was also significantly lower than that in normal controls [21]. Thus, leukemic cell release is best explained by disruption of the MBB barrier [22].

Normally, endothelial cells are in a quiescent state, and when tissue is injured or otherwise damaged, endothelial cells will form new blood vessels through a highly coordinated process of vessel sprouting called angiogenesis [23]. The steps of angiogenesis include detachment of existing vessels, degradation of the matrix, migration of endothelial cells, and the formation of a functional vascular plexus supported by pericytes and basement membrane components [24]. Tumor angiogenesis is the same process, but the proliferative activity of endothelial cells is significantly increased, and the vascular plexus also has remarkable functional and structural differences. In terms of ultrastructure, it is mainly manifested as many “openings” (endothelial fenestrations and transcellular pores) in the vessel wall, widened endothelial space, poor continuity, or absence of basement membrane, resulting in MBB loss (Figure 1) [25]. Structurally abnormal vessels affect local blood flow, metabolite exchange, and oxygenation, resulting in poor drug delivery and favoring the formation of a microenvironment for therapeutic resistance and recurrence [26].

**Table 1 diagnostics-14-00018-t001:** Cellular composition, function, and functional damage of the MBB.

Components	Location	Morphology of Cells	Function	Functional Damage
Endothelial cells	On the internal surface.	Thin and flat, the two ends of the cell are tapered and form the thin portion [9].	Regulate the number and timing of mature blood cells entering the blood circulation from the hematopoietic region [9].	Large gaps or damaged zones appeared in the endothelium and the continuity was poor [20].
Adventitial cells	On the outermost layer.	Showed a pyramidal shape whose top was protruded vertically into the myeloid tissue [9].	Regulate the coverage area and control the size of endothelial cell fenestra [10,11].	Adventitial cell cover rate was significantly reduced [21].
Basement membrane	On the medial surface.	Basement membrane in the bone marrow sinus often discontinuous [9].	Regulates the exchange and entry of chemicals and cells inside and outside the medullary sinus.	Basement membrane degeneration [20].

## 3. Assessment Methods of MBB

At present, the understanding of the MBB is based on SEM or TEM. With further study of the MBB, a variety of new imaging methods have been found to study the permeability of the MBB endothelial sinuses. A commonly used method is DCE-MRI, and the blood pool contrast agent can be used specifically to assess the permeability of the MBB. IVIM, ASL, and BOLD-MRI can indirectly reflect the permeability of the endothelial sinus wall of the MBB. As shown below (Figure 2), we present the raw and post-processing diagrams using different MRI techniques. In addition, the microfluidic device is the latest proposed method and is expected to be further developed and used to directly evaluate the degree of destruction of the MBB by cancer cells and the permeability of the MBB endothelial sinus wall.

### 3.1. Search Strategy

PubMed databases were searched and supplemented with a manual search. The following combination of keywords was used in the databases: marrow–blood barrier or endothelial barrier or blood–bone marrow barrier, hematologic diseases, functional magnetic resonance imaging or MRI, electron microscopy, angiogenesis. The search time was set from the establishment of the database to August 2023. Literature inclusion criteria were set as follows: (1) clinical or basic research on anatomical changes related to the MBB and angiogenesis in blood system diseases by magnetic resonance imaging, electron microscopy, and other techniques; (2) the type of literature was monographs, papers, reviews, or case reports. Exclusion criteria: (1) repeated articles; (2) the level of evidence was relatively insufficient and the research literature of low quality; (3) papers with low relevance to the topic; (4) the full text of the literature could not be obtained. A total of 2013 articles were retrieved; 285 duplicate articles were excluded, 915 articles were excluded by browsing the title and abstract, 553 articles were excluded according to the inclusion and exclusion criteria, 159 articles were excluded after browsing the full text, and finally 101 articles were included in this review.

### 3.2. DCE-MRI

#### 3.2.1. Imaging Methods That Need Small Molecular Contrast Agents

Although a variety of MRI techniques have been used to assess BM microcirculation flow, DCE-MRI is the most commonly used methods to assess microcirculation hemodynamics [27,28,29]. DCE-MR is used to evaluate the structure and function of micro-vessels by tracking the pharmacokinetic changes of small molecule contrast agents such as Gd-DTPA [30,31], which is a non-invasive examination method based upon an open two-compartment model [32]. Studies have shown that DCE-MRI can not only reflect the number of tumor blood vessels but also evaluate the permeability of tumor blood vessels, which has become a research hotspot in the field of tumor imaging [33].

Semiquantitative analysis is mainly used to obtain a group of parameters related to hemodynamics through a time-signal intensity curve (TIC), which is used to analyze the characteristics of tissue enhancement and indirectly reflect vascular permeability and tissue perfusion [34]. The commonly used semiquantitative parameters include peak, maximum enhancement (E_max_), time to peak (TTP), enhancement slope (E_slope_), blood flow, blood volume (BV), etc. The peak represents the concentration of contrast agent in both the intravascular and extravascular extracellular spaces, while the E_max_ value reflects the contrast concentration in the extracellular compartment, and the E_slope_ predominantly indicates the contrast concentration in the intravascular space [35]. Although semiquantitative analysis can directly reflect the inflow and outflow of contrast agents in tissues, it lacks a specific pharmacokinetic model and cannot accurately reflect the changes in the concentration of contrast agents in tissues. Quantitative analysis can directly reflect vascular permeability and tissue perfusion. The pharmacokinetic model (Tofts, Extended Tofts, or Brix model) was fitted with TIC to obtain the contrast agent concentration time curve to quantitatively evaluate the microvascular permeability and tissue perfusion [35]. Commonly used quantitative parameters include extravascular extracellular volume fraction (V_e_), volume transfer constant (K^trans^), amplitude (Amp), rate transfer coefficient (k_ep_), and elimination rate constant (K_el_) [36], where K^trans^ represents the rate of contrast agent penetration from the vascular lumen into the extravascular extracellular space, which is closely related to microvascular permeability and microvascular area. K_ep_ represents the contrast exchange between plasma and extravascular extracellular space [37], K_el_ value mainly depends on renal excretion, and Amp reflects the extravascular extracellular space of the contrast agent in the tissue [38,39].

#### 3.2.2. Clinical Applications

Leukemia

In the progression of AML, leukemia cells can infiltrate normal bone marrow and result in increased cellularity with more blasts and tumor angiogenesis [40]. DCE-MRI has also been applied to AML patients as a means of studying tumor angiogenesis. Studies have shown increased BM angiogenesis as measured by DCE-MRI in AML patients [41,42]. In addition, the peak enhancement rate of DCE-MRI reflects tissue perfusion, represents the sum of vascular density and permeability factors, and is an independent prognostic factor for overall survival (OS) in AML patients [42]. Studies have shown that the values of E_max_ and E_slope_ are positively correlated with the histological grade of BM infiltration, and the values of E_max_ and E_slope_ decreased and TTP increased in patients with effective treatment. This implies that the hemodynamic changes in BM microcirculation occurred at the same time as the remission of chemotherapy (the leukemic cells in bone marrow were less than 5%, and there were no immature cells in peripheral blood, and no extramedullary infiltration), which showed that the amount of tumor angiogenesis decreased and the permeability of blood vessels decreased [32]. DCE-MRI is also used to evaluate the therapeutic effect in patients with leukemia. In 51 AML patients who achieved complete remission, multiple DCE-MRI parameters were highly correlated with survival, including peak, E_slope_, Amp, and K_ep_ and K_el_. After achieving complete remission, the permeability of bone marrow vascular endothelium decreases, the tumor burden decreases, and the bone marrow perfusion decreases. Therefore, peak, E_slope_, Amp, and K_ep_ are negatively correlated with the survival time of patients. K_el_ value and the survival time of patients were positively correlated; among them, K_ep_ was an independent predictor of recurrence-free survival and OS [43].

2.Multiple Myeloma

Multiple myeloma is a hematological disorder characterized by the proliferation of plasma cells producing abnormal monoclonal immunoglobulin and infiltrating bone marrow [44,45]. Angiogenesis is an indicator of tumor growth, and total vessel area and micro-vessel density were increased in BM specimens from myeloma patients compared to monoclonal gammopathy of undetermined significance (MGUS) in undetermined significance patients, supporting the theory of angiogenesis [46,47,48] (MGUS is a group of diseases characterized by monoclonal plasma cell proliferation, which refers to the presence of M protein in serum, which can sometimes progress to MM). The DCE-MRI TIC of BM malignancies showed an earlier, steeper, and higher rise, reflecting the marked neovascularization of the tumor [49]. Moreover, semiquantitative parameters (peak, E_slope_) and quantitative parameters (Amp, K_el_) of DCE MRI were also demonstrated to be positively correlated with MVD [50]. DCE-MRI can also reliably assess the effect of treatment. Post-treatment responders had significantly lower amplitude and exchange rate constant k_ep_ values than non-responders, and higher post-treatment k_ep_ values were associated with shorter OS. Because treatment is accompanied by a reduction in bone marrow MVD and functional normalization of tumor vessels.

#### 3.2.3. Imaging Methods That Need Macromolecular Contrast Agents

During DCE-MRI imaging, small molecular contrast agents have high trans-endothelial permeability in both normal and tumor micro-vessels, and complex pharmacokinetic models still cannot accurately measure BV [51]. Compared with small molecule contrast agents, the distribution of macromolecular contrast media in healthy tissues is comparable to that of BVs, and its diffusion across micro-vessels is affected by endothelial permeability and has a longer plasma half-life. Abnormal trans-endothelial leakage is shown only in various pathological states of inflammation and malignancy [52,53,54,55,56]. Ultrasmall superparamagnetic iron oxide (USPIO, approximately 10–50 nm) is considered to be a specific macromolecular blood pool contrast agent. Because of its large size, it is mostly confined to the blood vessels of normal tissues after intravenous administration [57]. In normal hematopoietic BM, less than 5% of intravascular USPIO particles will extravasate through the MBB into the marrow interstitium [58,59]. Therefore, blood pool contrast agents can more accurately evaluate the local blood volume and also help to reflect abnormal vascular permeability. Currently, commonly used agents include ferumoxtran-10 (20–50 nm) and AMI-227 (17–20 nm), which can make the local magnetic field uneven, accelerate proton dephasing, and shorten the transverse relaxation time (T2WI) and longitudinal relaxation time (T1WI) of the tissue so that the corresponding T2WI and T2*WI tissue signals decrease and the T1WI tissue signals increase [60]. The BM signal intensity (SI) and signal-to-noise ratio (SNR) of pre-enhancement, post-enhancement, and dynamic images were measured. The ∆R2* map parameters of tumor perfusion patterns were analyzed, and the changes in the R2* relaxation rate (∆R2*) and vascular volume fraction (VVF) induced by iron oxide were calculated.

#### 3.2.4. Clinical Applications

Leukemia

Matuszewski showed that BM from AML patients had distinct areas of high vascularization with uneven vascular distribution on USPIO-induced ∆R2* maps, whereas BM from controls had areas of moderate vascularization with uniform vascular distribution. Quantitative analysis showed that VVF values were significantly higher in AML patients than in controls. The USPIO-induced ∆R2* changes and the extrapolated VVF correlated well with the MVD expression of the tumor [61]. Because the VVF is a surrogate marker for angiogenesis [62], USPIO-enhanced MRI can be applied to visualize and quantify BM infiltration.

2.Lymphomas

Compared with the untreated BM, Stephan found increased accumulation of ferumoxtran-10 in the BM of treated patients with non-Hodgkin’s lymphoma (NHL), manifested as a higher ΔSI after pretreatment compared with patients, which may be related to a treatment-induced increase in sinus endothelial permissiveness to the contrast agent [57]. Alterations of the sinus endothelium by condition therapy have been described previously and are mainly manifested as membrane shedding on the luminal surface, reduction of endothelial surface area, and loss of sinus endothelial integrity [63]. It is implied that MBB permeability can be assessed and quantified noninvasively in vivo using macromolecular contrast agents, such as ferumoxtran-10.

3.Total Body Irradiation

After total body irradiation, the sinusoids showed severe damage, the plasma membranes of sinus endothelial cells and hematopoietic cells in the sinus were sloughed, the endothelium lost its continuity, and the MBB became ineffective [64,65]. The effects of radiation on biofilms have been demonstrated in many other cellular systems [66,67]. In addition, histopathological studies have shown that the MBB fails after irradiation, and cell exchange can occur freely between the vascular and hematopoietic compartments. Daldrup’s study found that the same SI behavior was observed in irradiated BM compared to non-irradiated BM. However, the irradiated BM demonstrated an even higher AMI-227 enhancement. This tissue–blood relationship suggests that the injected contrast agent penetrates into the BM interstitium because of the discontinuity of the BM sinus endothelium [52].

#### 3.2.5. Advantages and Disadvantages

Small molecular contrast agents will quickly penetrate from the blood vessels to the outside of the blood vessels, and there is no unified pharmacokinetic model for different tissues. Macromolecular contrast agents can achieve long-term circulation (macromolecular contrast agents of high molecular weight prevent leakage to the stroma, and compared with the traditional drugs, they stay within the vascular system for longer periods of time), which is conducive to targeted imaging and high-resolution imaging, and it shows better biosafety. Unfortunately, these contrast agents sometimes produce a strong susceptibility effect known as “blooming” on MR images, which is manifested as a marked reduction in SI, leading to distortion or disappearance of organ boundaries [68].

The results of selected studies on the application of DCE-MRI in the diagnosis of hematological system diseases are shown in Table 2.

### 3.3. IVIM

Diffusion-weighted imaging (DWI) is a method for exploring tissue structures on the microscopic scale, which also plays an important role in non-solid tumors in addition to solid tumors [69]. IVIM is a DWI method using multiple b-values and a biexponential signal model [70,71,72], which can simultaneously obtain diffusion and perfusion quantification in a single image without an intravenous contrast agent [73]. The IVIM parameters include the pure molecular diffusion coefficient (*D*, indicating the tissue water diffusion coefficient and related to the structure of tissue cells), perfusion fraction (*f*, representing the vascular volume fraction), i.e., the fraction of the tissue that consists of vessel lumen, and perfusion-related diffusion coefficient (*D**, indicating the vascular flow rate) [74,75,76,77]. The normal vascular flow rate was *D** = 41.65 × 10^−3^ mm^2^/s in women and *D** = 95.07 × 10^−3^ mm^2^/s in men [78]. Although the pathophysiological significance of IVIM parameters needs to be further investigated, *f* and *D* values have been shown to be imaging markers for the independent assessment of vascular volume fraction for angiogenesis and cellularity in solid and hematologic tumors [79,80,81,82]. In recent years, to improve the effectiveness of IVIM parameters in characterizing specific tissues, a new IVIM parameter, the perfusion–diffusion ratio (PDR), has attracted attention. This parameter expresses the relationship between the rate of S(b) signal decline induced by IVIM and the rate of S(b) signal decline induced by diffusion. Therefore, IVIM is thought to provide information about MBB integrity [83].

#### 3.3.1. Clinical Applications

In acute leukemia, malignant clones of hematopoietic stem cells infiltrate normal BM, leading to cell proliferation accompanied by more blasts and tumor angiogenesis [84]. The results showed that the *f* value based on abnormal blood sinus endothelial cells in the lumbar BM microenvironment of AML patients was a prognostic factor for predicting the survival rate of AML patients. The *f* value of BM IVIM is positively correlated with histological MVD, and the *f* value can reflect the difference in blood perfusion between acute lymphoblastic leukemia (ALL) and AML. Therefore, the *f* value can be used as a surrogate imaging marker of AL angiogenesis [85]. However, AML patients will have changes such as a decrease in sinus diameter and an increase in sinus endothelial permeability. Whether the *f* value can accurately reflect the change in BV caused by the increase in sinus endothelium needs further study [86]. The *D* value more accurately reflects the cell density. Lower *D* values were associated with higher cell proliferation and tumor burden, and thus lower *D* values were associated with shorter OS. As another perfusion parameter in the IVIM model, *D** mainly reflects the capillary blood flow rate. Its limitation is that the SNR is poor compared with the other two parameters. The study showed that there was no statistically significant correlation between lumbar BM *D** and OS in untreated patients, but the survival rate of patients with *D** < 115 × 10^−3^ mm^2^/s was lower than that of patients with *D** > 115 × 10^−3^ mm^2^/s and higher-than-normal values. The increased BM vascular permeability in AML patients is a poor prognostic factor, so it is reasonable to assume that increased vascular permeability causes decreased blood flow velocity, which is consistent with lower survival in patients with lower *D** values [73].

#### 3.3.2. Advantages and Disadvantages

As a non-contrast perfusion imaging modality, IVIM can be used in some patients with contraindications to contrast media, such as severe renal impairment [73]. In addition, IVIM diffusion MR imaging does not involve the injection of ionizing radiation or radioisotopes. However, the IVIM parameter values are affected by many factors, such as respiratory motion artifacts and blood flow artifacts. In addition, the sensitivity of IVIM MRI varies not only by vessel size but also by the number of b values applied.

Table 3 lists some of the diagnostic applications of IVIM techniques in hematological diseases.

### 3.4. ASL

ASL, as a noninvasive and non-contrast-enhanced perfusion imaging method [87], provides important functional information on tumor proliferation and tumor perfusion as a surrogate marker for angiogenesis and MVD [88]. Unlike DCE-MRI, ASL imaging uses endogenous blood as an endogenous reagent to measure tissue perfusion [89]. The tracer consists of blood water in the artery, which is generated by inverting the magnetization of the blood in the arteries that feed the organ of interest with a train of radio frequency pulses [90]. When the labeled water proton flows into the imaging plane with blood flow, the longitudinal relaxation T1 of the local tissue changes, so the labeled image is obtained. Then, unlabeled blood is imaged once at the same level with the same parameters, which acts as the control image. Subtraction of the labeled image from the control image results in an image containing only perfusion information based on the information obtained from the perfusion imaging (such as signal intensity change) [91]. Encouragingly, ASL imaging may have the potential to detect changes in BM perfusion associated with the diagnosis of lesions [87].

#### 3.4.1. Clinical Applications

The primary means of assessing the course of disease in patients with myeloma remains hematological parameters. However, as noted by Salmon and Durie in 1975, serum paraprotein levels do not always adequately reflect tumor burden [92]. In recent years, modern MR techniques have been developed to help characterize and follow up on solid and non-solid tumors. Fenchel performed quantitative perfusion measurements in myeloma with the use of a flow-sensitive alternating inversion recovery (FAIR) perfusion sequence, with MR perfusion techniques used to provide the information of microvascular tissue perfusion. After the therapy began, microvascular perfusion could be reduced, so the effective treatment in patients with MM showed significantly decreased perfusion, and treatment with responder baseline perfusion value was higher than no reaction [88]. The results of the Schor-Bardach study show that contrast-free ASL MRI can directly assess neovascularization, and has a good correlation with MVD [93].

#### 3.4.2. Advantages and Disadvantages

ASL imaging is a non-radioactive, non-contrast-enhanced, non-invasive method that is well suited for longitudinal monitoring of disease progression and routine assessment of treatment response, especially in patients with renal insufficiency and children [87]. Its disadvantages are temporal and spatial resolution and low signal-to-noise ratio.

### 3.5. BOLD-MRI

BOLD-MRI uses blood as an endogenous contrast agent and primarily detects changes in the intravascular hemoglobin ratio. Oxyhemoglobin is a diamagnetic substance, and deoxygenated hemoglobin is a paramagnetic substance. When the deoxygenated hemoglobin/oxyhemoglobin ratio increases, the magnetic field of the surrounding local tissue will be uneven, resulting in proton spin de-phase, shortening the transverse magnetization T2*, and a low T2*WI signal [94]. Duan found that the BOLD signal fluctuation ability in the center of malignant tumors was lower than that in the surrounding area, while this difference was not observed in benign tumors. Such BOLD signal fluctuations may be related to acute hypoxia in the tumor [95]. A study shows that BM hypoxia is significantly increased after AML engraftment compared with normal human engraftment, and in the early stage of transplantation, BM hypoxia is mainly concentrated near AML cells, while the whole BM is hypoxic at the time of high transplantation because AML engraftment not only induces angiogenesis within the BM but also alters the normal vascular structure and function [96]. These new abnormal vessels are unable to provide nutrients and oxygen (as well as drugs) uniformly in the BM tissue and may help maintain overall hypoxia [26]. In the future, it is hoped that we can apply this technique to evaluate the permeability of the MBB.

#### 3.5.1. Advantages and Disadvantages

As a non-invasive and safe MRI technique, BOLD-MRI is sensitive to changes in blood supply, vascular performance, and cell function. BOLD-MRI signal is an indirect indicator of tissue oxygenation. There is a lack of effective calibration methods to quantify R2* value and tissue oxygen partial pressure, so it is difficult to quantitatively evaluate oxygen content [97].

### 3.6. Other

In recent years, significant progress has been made in the development of micro-structured cell culture devices that reproduce organ physiological function in vitro and mimic the pathophysiological development of human diseases [98]. Organ chips simulate the physiological function of tissue cells via spatiotemporal control of organ-specific biological, chemical, and physical factors in the tissue microenvironment. Novel microfluidic culture devices can be used to study cell migration in the sinusoidal microenvironment of BM. Methods: (1) the blood circulation in the BM, represented by the flow of culture medium in the sinus cavity with physiological τ_w_ of ~0.1 Pa; (2) the BM sinusoidal endothelium, represented by monolayer endothelial cells; and (3) the BM stroma, represented by fibroblasts with a collagen fibroblast-like distribution in the interstitial compartment [99].

#### 3.6.1. Clinical Applications

Chao’s study found that the egress of multiple myeloma cells resulted in a poor organizational structure of endothelial cells, loose junctions, widening of endothelial cell junctional pores, and increased permeability through endothelial cells. These endothelial cell morphological changes appear to mimic the abnormal angiogenesis observed in sinusoidal endothelial cells during the progression of active multiple myeloma [100,101]. Compared with observation in vivo, the study also found that the marrow of sinus endothelial pore and barrier function can also be reproduced in the device, so the device can be used to reconstruct the endothelium and BM matrix and to assess the barrier function of the endothelium by measuring permeability.

#### 3.6.2. Advantages and Disadvantages

Compared with traditional molecular diagnostic techniques, microfluidic devices are highly integrated, small, and accurate, but their consumables are expensive and the demand is small.

The summary of the results of selected studies in the use of ASL, BOLD-MRI, and microfluidic devices for hematological system diseases are shown in Table 4.

## 4. Conclusions

Hematological malignancies are malignant tumors originating from the blood system, and their occurrence and development are related to the abnormal proliferation and differentiation of the hematopoietic system. Although new therapies such as bone marrow transplantation and anti-tumor vascular therapy have improved the curative effect of the disease, the mortality rate remains high. Individualized treatment based on the MBB may enable patients to achieve the greatest possibility of a cure. But our understanding of the MBB and its associated disorders is still in its infancy. The continuous development of bone marrow imaging techniques has provided new insights into the underlying tumor biology, allowing us to transition our understanding of the MBB from the descriptive stage of morphological analysis to the analytical stage of functional imaging. This process needs to clarify the relationship between various functional imaging modalities and the MBB, which will also help to better understand the biological characteristics of the MBB and open up new methods for the diagnosis and prognosis of the disease. At present, the diagnosis of hematological malignancies is mainly based on histopathological analysis of bone marrow biopsy specimens, which is invasive and limits its wide clinical application. In addition, local specimens cannot dynamically and comprehensively reflect tumor heterogeneity. Advanced MRI functional imaging technology can reflect the changes of lesions from different angles (cell proliferation, blood perfusion, etc.). DCE-MRI, a non-invasive imaging method, is commonly used to assess the permeability of the MBB and analyze the degree of progression in related diseases, but it requires the injection of contrast media and is not suitable for patients with contraindications to contrast media. ASL and IVIM could noninvasively assess MBB permeability by adopting endogenous contrast, which would be widely used in clinical research. BOLD-MRI and microfluidic devices are rarely used in the blood system at present, but both of them have broad promise for the detection of MBB integrity in the future. This article summarizes the significance and evaluation methods of the MBB in hematological diseases, which may not be comprehensive, and some of the diagnostic techniques in this article are in the exploratory stage in hematological diseases. A large number of prospective studies and validation studies are needed to further provide data and theoretical basis for evaluating the function of the MBB. It is hoped to be able to further integrate MBB morphological and functional imaging studies to lay a theoretical foundation for the establishment of a prognosis evaluation system of hematological tumors based on MRI functional imaging, and to provide an imaging basis for formulating accurate personalized treatment plans for patients. Table 5 lists the current MRI techniques used to assess the MBB and their advantages and limitations.

## Figures and Tables

**Figure 1 diagnostics-14-00018-f001:**
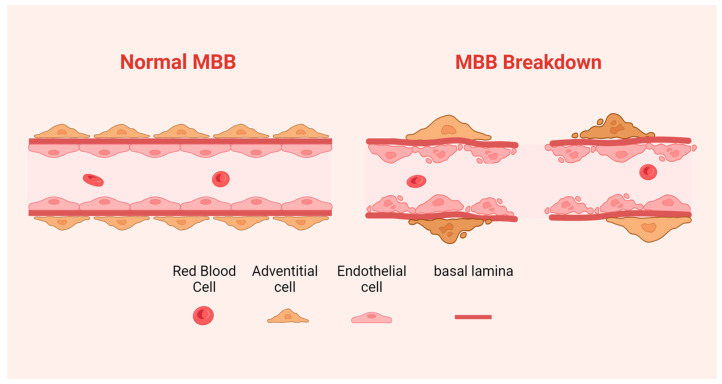
Schematic representation of normal MBB versus structurally damaged MBB.

**Figure 2 diagnostics-14-00018-f002:**
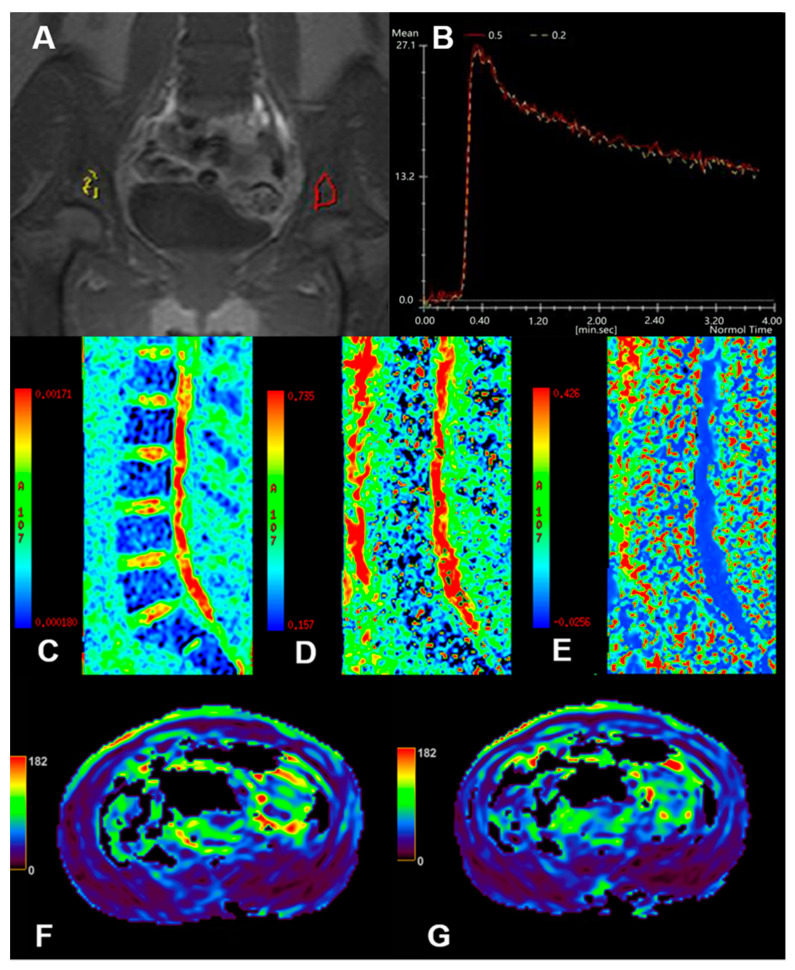
This picture shows partial raw and post-processed MRI maps of this review. Raw DCE-MRI image of leukemia bone marrow (**A**), two regions of interest (ROI) were set at the bilateral ilium, one in yellow and the other in red, and the TIC curve (**B**) showed a rapid increase and decrease type; IVIM *D* map (**C**), IVIM *f* map (**D**), and IVIM *D** map (**E**) of leukemia bone marrow; ASL perfusion map of leukemic bone marrow (**F**,**G**).

**Table 2 diagnostics-14-00018-t002:** A summary of selected studies on the application of DCE-MRI in the diagnosis of hematological system diseases.

Authors	Year	Modality Imaging	Subjects	Tracer	Parameter	Associated with MVD	Findings
Zha et al. [32]	2010	DCE-MRI	6 MM patients, 11 AL patients, 7 chronic myeloid leukemia patients, and 2 NHL patients vs. 20 normal volunteers.	Gadolinium	E_max_, E_slope_, and TTP	-	E_max_, E_slope_, and TTP can reflect the malignancies’ histological grade.
Shih et al. [41]	2006	DCE-MRI	17 AML patients vs. 17 normal volunteers.	Gadopentetate dimeglumine	E_max_, E_slope_	-	AML patients with high slope and high peak had significantly decreased OS.
Shih et al. [42]	2009	DCE-MRI	78 AML patients.	Gadopentetate dimeglumine	Peak, Amp, K^trans^	-	Peak and Amp can predict OS with AML patients.
Chen et al. [43]	2011	DCE-MRI	51 pretreatment AML patients.	Gadodiamide	Peak, E_slope_, Amp, and K_ep_	-	High values of peak, slope, amplitude, and Kep were associated with shorter OS.
Huang et al. [50]	2012	DCE-MRI	49 MM patients.	Gadodiamide	Peak, E_slope_, Amp, K_ep_, and K_el_	Apart from Kep, all parameters correlate positively with MVD.	High Amp values as a possible risk factor associated with the development of extramedullary disease in MM patients.
Daldrup et al. [52]	2000	DCE-MRI	20 6-week-old female New Zealand white rabbits.	CMD-Gd-DOTA	ΔSI	-	Using CMD-Gd-DOTA-enhanced MRI, the integrity of the blood–bone marrow barrier could be accurately defined in vivo.
Stephan et al. [57]	2006	DCE-MRI	22 NHL patients.	Ferumoxtran-10	ΔSI	-	Increased ferumoxtran-10 accumulation correlated with alterations of the MBB.
Matuszewski et al. [61]	2007	DCE-MRI	11 AML patients vs. 6 normal volunteers.	Iron oxide blood pool contrast agent	R2*, VVF	∆R2*and VVF correlate positively with MVD.	Blood pool contrast agents can noninvasively image tissue angiogenesis.

**Table 3 diagnostics-14-00018-t003:** Partial applications of IVIM technique in the diagnosis of hematological diseases.

Authors	Year	Modality Imaging	Subjects	Tracer	Parameter	Associated with MVD	Findings
Fan et al. [68]	2020	IVIM	15 Anemia patients vs. 28 AL patients.	-	*f*, *D*, *D**	*f* value is positively correlated with the histological features of marrow.	MVD and *f* values of AL were higher than those of anemia.
Li et al. [73]	2020	IVIM	53 AML patients.	-	*f*, *D*, *D**	-	Higher *f* value and lower *D* value were indicative of unfavorable OS; *f* value was shown as an independent prognostic factor.
Niu et al. [79]	2017	IVIM	53 AML patients.	-	*D*, *D**, *f*, and ADC	-	*D* and *f* values can be used to predict the treatment response of AML patients.
Bourillon et al. [80]	2015	IVIM/DCE-MRI	27 MM patients.	-/Gadolinium chelate	*f*, *D*, *D**, ADC, E_max_	-	*D* and ADC values positively correlated with E_max_.
Li et al. [85]	2020	IVIM	20 AL patients.	*-*	*f*, *D*, *D**	*f* value is positively correlated with the histological MVD.	IVIM parameters combined with histological parameters can evaluate the microstructure of AL bone marrow.

**Table 4 diagnostics-14-00018-t004:** Summary of selected studies on the application of ASL, BOLD-MRI, and microfluidic device in the diagnosis of hematological system diseases.

Authors	Year	Modality Imaging	Subjects	Tracer	Parameter	Associated with MVD	Findings
Li et al. [87]	2020	ASL	8 healthy volunteers vs. 2 juvenile osteochondritis dissecans.	-	Perfusion signal.	-	ASL imaging can detect disease-related bone marrow perfusion changes.
Fenchel et al. [88]	2010	ASL	19 MM patients.	-	Perfusion signal.	-	Correlation of ASL perfusion with outcome was significant.
Duan et al. [95]	2016	BOLD-MRI	52 musculoskeletal tumor patients.	-	Frequency band.	-	BOLD signal is positively related to blood flow or the partial pressure of oxygen in normal tissue or tumor tissue.
Sui. et al. [99]	2022	Microfluidic devices	-	-	-	-	Microfluidic devices can be used to study the trafficking of MM cells through the sinusoidal niche of the BM.
Ferrarini. et al. [101]	2013	3-D culture models	5 MM patients vs. 4 joint replacement patients.	-	-	-	3-D culture models can assess the metabolic activity of MM cells and their microenvironment.

**Table 5 diagnostics-14-00018-t005:** A summary of the assessment methods of the MBB.

Methods	Principle	Advantages	Limitations	Previous Study and Clinical Significance
DCE-MRI	Evaluates the structure and function of micro-vessels by tracking the pharmacokinetic changes of molecular contrast agents [30,31].	Imaging methods that require small molecular contrast agents are non-invasive examination methods [32].	Have high trans-endothelial permeability in both normal and tumor micro-vessels and there is no unified pharmacokinetic model [51]; longer scan time.	Shih. et al. [42] Increased bone marrow angiogenesis measured by DCE-MRI can predict adverse clinical outcome in AML patients. Chen. et al. [43] Bone marrow perfusion measured with DCE-MRI in AML patients in CR can be an indicator of outcome and survival. Moulopoulos. et al. [49] It may identify malignant bone marrow infiltration in patients with negative static MRI and serve as a diagnostic tool for patients with bone marrow malignancies. Huang. et al. [50] Amp correlated strongly with MVD in BM and also extramedullary disease in patients with MM.
		The distribution of macromolecular contrast media in healthy tissues is comparable to that of BVs and has a longer plasma half-life [52,53,54,55,56].	Sometimes produce a strong susceptibility effect that causes organ boundaries to distort or disappear [69]; decreased temporal resolution.	Matuszewski. et al. [61] The USPIO-induced ∆R2* changes and the extrapolated VVF correlated well with the MVD expression of the tumor; can be applied to visualize and quantify BM infiltration. Metz. et al. [57] Macromolecular contrast agents can non-invasively assess and quantify the permeability of MBB in vivo. Daldrup. et al. [52] Irradiation-induced alterations in MBB permeability could be reliably assessed with dynamic MRI, using the new macromolecular contrast agent CMD-Gd-DOTA.
IVIM	Using multiple b-values and a biexponential signal model to obtain information on the diffusion of tissues and capillary perfusion [70,71,72].	Simultaneously obtain diffusion and perfusion quantification in a single image without an intravenous contrast agent [73].	The parameter values are affected by many factors, such as respiratory motion artifacts; parameter values are highly variable; susceptibility artifacts.	Li. et al. [85] The *f* value derived from IVIM in bone marrow was positively correlated with MVD, and can be used as an alternative imaging marker of angiogenesis in marrow of AL patients. Niu. et al. [79] The *f* value based on abnormal blood sinus endothelial cells in the lumbar BM microenvironment of AML patients was a prognostic factor for predicting the survival rate of AML patients. Li. et al. [73] Decreased blood flow rate in AML patients may be related to increased vascular permeability, and *D** value may be used to evaluate bone marrow vascular permeability in AML.
ASL	Using endogenous blood as reagent, the perfusion image was obtained with a subtraction technique [91].	Noninvasive and non-contrast-enhanced perfusion imaging method that is highly reproducible [87].	Relatively poor parameters, time and space resolution, and low SNR.	Fenchel. et al. [88] ASL MRI can reliably detect early response to anti-angiogenic therapy in patients with multiple myeloma. Bardach. et al. [93] ASL MRI can reliably assess BM infiltration and neovascularization, and has a good correlation with MVD.
BOLD-MRI	Using endogenous deoxygenated hemoglobin to transform microscopic magnetic fields and de-phase spin protons [94,95].	Non-invasive and non-contrast-enhanced perfusion imaging method [94,95].	Lack of unified technical standards at home and abroad, and lack of effective calibration method.	Duan. et al. [95] BOLD signal fluctuations may be related to acute hypoxia in the tumor.
Microfluidic Devices	Simulate the physiological function of tissue cells with organ-specific biological, chemical, and physical factors [98].	Can directly observe the changes of tissues in various states; small and accurate. [99,100].	Consumables are expensive and the demand is small.	Sui. et al. [99] The device can be used to study the physical and secreted factors determining the trafficking of cancer cells through BM. Ferrarini. et al. [101] A 3-D, RCCS™ bioreactor-based culture of tissue explants can be exploited for studying myeloma biology and for a pre-clinical approach to patient-targeted therapy.

## Data Availability

Not applicable.

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
