# Peer review of "Imaging Techniques and Clinical Application of the Marrow–Blood Barrier in Hematological Malignancies"

_diagnostics, 2023, doi:10.3390/diagnostics14010018_

Round 1

Reviewer 1 Report (Previous Reviewer 1)

Comments and Suggestions for Authors

The authors have made significant improvement in this review paper and added more relevant details in the tables. They are now more informative and better organized. 

Author Response

       Thank you very much for taking the time to review my manuscript and providing valuable feedback and suggestions. I really appreciate your comments and suggestions,  thank you again for your review and support.

Reviewer 2 Report (New Reviewer)

Comments and Suggestions for Authors

The article presents a valuable contribution to the field by reviewing various imaging modalities for assessing the MBB. 

The introduction provides a foundational understanding of the significance of assessing the Bone Marrow Blood Barrier (MBB). Strengths include a clear presentation of the importance of MBB assessment and a succinct problem statement. May consider expanding the introduction slightly to include a brief background or context for readers who may not be familiar with MBB and its relevance in medical imaging.

The methods section outlines the criteria for selecting studies, but there is room for improvement in terms of transparency and clarity. Specify inclusion and exclusion criteria for selecting studies, ensuring transparency in the review process.

The conclusion summarizes the findings but could benefit from explicitly stating the implications and linking back to the initial problem statement. Additionally, acknowledging limitations and proposing future directions for research would strengthen this section.

Author Response

       Thank you very much for taking the time to review my manuscript and providing valuable feedback and suggestions. I really appreciate your comments and suggestions, and have made modifications and improvements based on your feedback. For details, please check the attachment, thank you again for your review and support.

Reviewer 3 Report (New Reviewer)

Comments and Suggestions for Authors

This review describes imaging and clinical applications of the marrow-blood barrier (MBB) in hematological cancers. In particular, the MBB is often severely disrupted in hematological malignancies. Consequently, proper imaging in these unusual and challenging cases will significantly contribute to understanding and proper treatment of these patients. Accordingly, this review makes an important contribution to the field by introducing MBB imaging modalities along with their clinical applications. Overall, this Review is well-organized and illustrated. Several tables nicely summarize important information. The Review is also reasonably thorough but not excessively long. 

Author Response

Thank you very much for taking the time to review my manuscript and providing valuable feedback and suggestions. I really appreciate your comments and suggestions, thank you again for your review and support.

Reviewer 4 Report (New Reviewer)

Comments and Suggestions for Authors

This is an interesting review of novel MBB assessment methods in common hematological malignancies and the clinical relevance and application of each approach. It describes the methods rather comprehensively but at times can be repetitive and difficult to follow with the use of terminology such as naive leukocyte hemogram, long-term circulation, and remission of chemotherapy. In the discussion on clinical applications of DCE-MRI in leukemia, details on how this can be used to evaluate clinical response would be useful in helping the reader understand- for example which parameters are correlated with improved survival and why so. Similarly, the section on advantages/disadvantages (3.1.3) - the first paragraph doesn't make sense.

Overall, an interesting paper which can be improved with more relevant details relating to clinical applications in its attempts to increase understanding of how these methods can be used and the limitations in these techniques.

Comments on the Quality of English Language

Adequate with some difficult to understand terminology used- as mentioned above

Author Response

Thank you very much for taking the time to review my manuscript and providing valuable feedback and suggestions. I really appreciate your comments and suggestions, and have made modifications and improvements based on your feedback. For details, please check the attachment, thank you again for your review and support.

This manuscript is a resubmission of an earlier submission. The following is a list of the peer review reports and author responses from that submission.

Round 1

Reviewer 1 Report

Comments and Suggestions for Authors

This is a review of an interesting topic. However, the delivery was unfortunately lacking, with lack of good organization and presentation of data. There were things that could've been presented better in a table format (e.g. results of previous study, clinical significance, etc). Only hematological conditions were elaborated while the techniques applied to solid tumors as well. 

Comments on the Quality of English Language

There are many grammatical / syntax errors, and failure to use of capital letters, which made the manuscript difficult to read. 

Reviewer 2 Report

Comments and Suggestions for Authors

The manuscript provides a comprehensive overview of the Marrow-Blood Barrier (MBB), its structure, function, and assessment methods.
   The link between MBB dysfunction and diseases, like acute leukemia and preleukemia, is well established. The focus on imaging methods is relevant and essential for understanding the role of MBB in health and disease.    The writing is clear and accessible, making the complex subject matter understandable to a broad audience.

Me recommendations:
   Please add relevant citations in table 1 and 2, which is crucial to support the information provided.

   Some sentences could be revised for better readability and flow. A careful review for minor grammatical errors is needed.
For example  line 91 (physiological molecules), 102 (unlimited proliferation) etc.  

   While the use of technical language is appropriate for the audience, providing brief explanations or examples could make the content more accessible to readers less familiar with the topic.

   In the final section should more comprehensively expand and summarize key takeaways, potential clinical applications, and areas for future research to give readers a complete perspective on the topic.

Comments on the Quality of English Language

Some grammatical errors should be avoided.

Reviewer 3 Report

Comments and Suggestions for Authors

The review is a complete analysis of the field and evaluates the pros and cons of the different techiniques and approaches suggested. The reading of the manuscript is easy and fit also for a non specialized audience.  

Author Response

I am very grateful to the reviewer for careful review and guidance of my article.